# Bessel Beam Femtosecond Laser Interaction with Fused Silica Before and After Chemical Etching: Comparison of Single Pulse, MHz-Burst, and GHz-Burst

**DOI:** 10.3390/mi15111313

**Published:** 2024-10-29

**Authors:** Théo Guilberteau, Pierre Balage, Manon Lafargue, John Lopez, Laura Gemini, Inka Manek-Hönninger

**Affiliations:** 1Université de Bordeaux-CNRS-CEA, CELIA UMR 5107, 33405 Talence, France; 2ALPhANOV, Rue François Mitterrand, 33400 Talence, France; 3Amplitude, Cité de la Photonique, 11 Avenue de Canteranne, 33600 Pessac, France

**Keywords:** Bessel beam, GHz-burst mode, laser processing, dielectrics, chemical etching, laser drilling, fused silica

## Abstract

We investigate the elongated modifications resulting from a Bessel beam-shaped femtosecond laser in fused silica under three different operation modes, i.e., the single-pulse, MHz-burst, and GHz-burst regimes. The single-pulse and MHz-burst regimes show rather similar behavior in glass, featuring elongated and slightly tapered modifications. Subsequent etching with Potassium Hydroxide exhibits an etching rate and selectivity of up to 606 μm/h and 2103:1 in single-pulse operation and up to 322 μm/h and 2230:1 in the MHz-burst regime, respectively. Interestingly, in the GHz-burst mode, modification by a single burst of 50 pulses forms a taper-free hole without any etching. This constitutes a significant result paving the way for chemical-free, on-the-fly drilling of high aspect-ratio holes in glass.

## 1. Introduction

Precise micro-machining of dielectric transparent materials, such as glass, has seen remarkable advancements over the past few decades, largely driven by the growing demand for high-precision techniques in industries like microelectronics, optics, and biomedical devices [1]. Early glass micromachining methods, such as chemical etching (e.g., wet etching, deep reactive ion etching, or mechanical machining), faced challenges related to low precision, material damage, and contamination [2]. The introduction of laser-based micromachining revolutionized the field by offering a contact-free and highly precise method for material removal. However, early continuous-wave (CW) and nanosecond (ns) laser systems still encountered issues, such as excessive thermal loads and residual stress, leading to cracking and reduced material integrity due to long interaction times and high-temperature gradients.

In the past decade, the field has witnessed significant progress with the development of industrial high-power femtosecond (fs) laser sources, enabling unprecedented precision and efficiency. Modern fs lasers, with pulse durations on the order of $10^{-15}$ s, employ non-linear absorption processes that localize the energy deposition near the focal plane, minimizing thermal damage and eliminating the mechanical stress associated with longer pulse systems [3,4]. This has made fs laser micromachining the method of choice for brittle, heat-sensitive materials like glass, where minimal thermal load and high accuracy are critical. Continuous advancements in fs laser technology have brought innovations in temporal beam shaping, improving control over various material modification processes.

More recently, researchers have explored advanced processing regimes, including MHz-burst [5] and GHz-burst modes [6,7,8], which involve delivering pulse trains with a high repetition rate within the burst. These burst modes offer significant advantages over the traditional single-pulse regime by reducing detrimental non-linear propagation effects upstream the focus point, such as self-focusing or intensity clamping, and by enhancing the deposited energy density [9] and improving material removal rates through controlled thermal accumulation [10,11,12,13]. When combined with Bessel beam shaping [14,15,16,17], these burst modes provide a powerful tool for high-aspect-ratio drilling [18] and cutting in transparent dielectrics like glass [19,20,21].

Femtosecond laser micromachining, especially with Bessel beam shaping, has since established itself as one of the leading techniques for microstructuring glass and other transparent materials [22,23,24,25]. The ability to create high-precision features with minimal thermal damage has expanded its applications in microfluidics, optics, and integrated photonic circuits [26,27].

The combination of fs laser glass processing and chemical etching has been widely explored in previous studies [28], particularly in crystalline materials, where anisotropic etching due to crystal orientation plays a significant role [29]. However, in amorphous materials like fused silica, chemical etching behavior is driven more by laser-induced modifications than by intrinsic material properties. Recent studies have demonstrated that Potassium Hydroxide (KOH) etching can effectively complement laser processing by enhancing the removal of modified regions in microchannel fabrication [30,31]. The high etching selectivity, which enables discrimination between treated and untreated areas, is critical in achieving high-aspect-ratio structures and smooth surface finishes.

In this study, we aim to use spatial Bessel-beam shaping to produce high-aspect-ratio modifications and to evaluate the influence of temporal beam-shaping in two different burst regimes (MHz and GHz), in comparison to the repetitive single-pulse regime (SP), (i) on the final hole shape and inner wall quality before and after chemical etching, and (ii) on the etching rate for enhanced drilling accuracy and compatibility with the etching technique [26,32]. Our findings constitute a significant contribution to advancing microstructuring techniques in transparent dielectrics, especially for applications requiring high-precision material removal. Moreover, we demonstrate direct fine-hole drilling with a single GHz-burst without any post-processing, which is extremely interesting for applications like Through Glass Via (TGV) fabrication with high aspect ratio.

## 2. Materials and Methods

### 2.1. Laser System

The laser system used for this experiment is a Tangor 100 from Amplitude (Amplitude Laser Group, Pessac, France), which emits pulses at 1030 nm wavelength with a pulse duration of 515 fs. The output beam is linearly polarized. This laser system can operate in repetitive single pulse (SP) mode, in MHz-burst mode with 4 pulses per burst (ppb) at 40 MHz pulse repetition rate and in GHz-burst mode with 50 ppb at an intra-burst repetition rate of 1.28 GHz. The burst repetition rate is adjustable up to 200 kHz and the pulse/burst energy available on the target ranged from 0 to 255 µJ. The measured pulse/burst shapes in these three regimes as in [33] are depicted in Figure 1.

### 2.2. Machining Workstation

A 2x beam expander is used to set the beam diameter to 10 mm at the entrance of the Bessel beam shaping module. The latter includes a 170° apex-angle axicon to generate the primary Bessel beam and a set of lenses $f_{1}=125$ mm and $f_{2}=10$ mm to demagnify this Bessel beam into a secondary Bessel beam with a half-angle of 28.6°. The theoretical non-diffracting zone length was calculated to 1.8 mm [19]. This setup was placed on a motorized Z-axis stage (Alio Industries, AI-LM-10000-I-PLT-LP) for precise positioning of the Bessel beam within the sample. Additionally, the sample was set on XY-monolithic motorized stages (Alio Industries, AI-LM-20000-XY-I-LP) to position it with respect to the Bessel beam and to control the translation for the different series of experiments. Moreover, the station is equipped with a side-view imaging system featuring a green diode emitting at 523 nm and a long-distance microscope (InfiniMax KX with MX5 objective) coupled with a Basler CMOS camera (Basler acA1440-220uc, resolution of 1440 × 1080, pixel size 3.45 μm × 3.45 μm, 227 fps, global shutter). Additionally, a 520 nm bandpass filter was used in order to avoid for the camera to be blinded by the processing light. The entire machining station is mounted on a stiff granite base and gantry to ensure experimental stability and repeatability. The translation stages, laser gate, image recording system, and power modulator unit, composed of a half wave plate and a beam splitter, are automated and managed through DMCpro software (Direct Machining Control, Vilnius, Lithuania), giving us complete control and repeatability over the machining station and its parameters. Figure 2 shows a schematic drawing of the experimental setup.

### 2.3. Etching After Laser Irradiation

Fused silica samples, readily available from Altechna (JGS1 22 × 30 × 1; 80-50 S-D), were selected for this study. Both side surfaces were polished to facilitate post-mortem side observation. After laser processing, the samples were immersed in a 10 mol·$L^{-1}$ KOH etchant solution, heated in an ultrasonic bath at 55 °C to ensure low temperature deviation during the etching time. The etching process was carried out for durations of 30 min, 1 h, and 2 h. Following the etching, the samples were cleaned in acetone for 5 min.

## 3. Results and Discussion

In this section, we first examine and compare the modifications induced by laser irradiation across the three regimes within the bulk material using a single pulse or burst, as detailed in Section 3.1. Subsequently, the evolution of the etching rate with respect to different etching times is presented together with a detailed analysis of the etching-induced bulk modifications in Section 3.2. Further, we compare the etching depths between the SP and MHz-burst regimes. Finally, we show that the GHz-burst regime allows for direct hole formation with a single burst, constituting a remarkable result paving the way for chemical-free and single-step high-aspect-ratio TGV fabrication (in Section 3.3).

### 3.1. Single Pulse/MHz-Burst/GHz-Burst Comparison

In the upcoming section, the modification depths after laser irradiation in fused silica will be presented as a function of the energy contained in a pulse or burst ranging from 15 µJ to 255 µJ. Each modification was repeated four times on the same sample with a pitch of 50 µm to ensure reliability of the results. We experimentally observed with our side-view system that the modifications resulting from a low-energy Bessel beam start from the middle and then expand to the top and rear surfaces, respectively. For the sake of repeatability, the Bessel beam was initially fully placed inside the medium, generating a modification with 255 µJ. Subsequently, by moving the Z-stage in the direction of the irradiation axis, 30% of the Bessel beam was placed above the top surface of the sample, and only 70% of the maximum interaction length was used to produce the modifications inside the sample. This placement, for all experimental series, guarantees that the modifications reach the top surface for the subsequent etching step, as described in Section 2.3. Moreover, the rising front of the Bessel beam [34] does not interact with the material avoiding intensity variations. Note that the Z-stage was kept in this position and not moved during the experiments. The resulting modifications for the single-pulse, MHz-burst, and GHz-burst regimes are depicted in Figure 3.

The images in Figure 3 were acquired with an in situ long-distance microscope equipped with a 1.4× zoom lens and are created by stitching multiple pictures together. The inserts on the right were taken with an optical measuring microscope (MF-B1010D, Mitutoyo) and a 50× objective. In Figure 3a and Figure 3b, respectively, the SP and MHz-burst regimes, for the range of investigated energy, all energies resulted in creating a visible modification in the sample. From Figure 3a, it can be observed that as the energy increases, the depth of the modification increases and becomes darker in the images, suggesting a stronger modification of the material. The zoomed images, specifically Figure 3a and Figure 3b, show detailed views of the modifications induced at the highest SP or burst energies, respectively. These modifications appear thin and elongated, with small bubbles along the modifications and a slightly tapered shape for the SP regime. For the MHz-burst regime, we observe similarly elongated modifications, but with bubbles that seem larger with a tapered shape. The size of the bubbles was not measurable with our optical microscope. In contrast, in Figure 3c, depicting the GHz-burst regime, we observe a sudden onset of energy resulting in modifications. The modifications observed are reproducible from 165 µJ burst energy and above. Below this threshold energy, we can observe that the modifications resulting from a single GHz-burst are very weak, not consistently reproducible and do not reach the surface. Applying more than one burst can produce results comparable to using higher energies with modifications reaching the top surface but this will not be discussed in this study. As the pulse energy is distributed into a burst of 50 pulses (2 µJ–5.1 µJ per sub-pulse were tested), non-linear absorption requires a higher intensity value. From 165 µJ to 225 µJ, we observe consistent modifications with increasing depth as a function of burst energy. In Figure 3c, the hole morphology corresponding to a burst energy of 255 µJ in GHz-burst mode is represented, we observe a taper-free, straight canal. However, it must be noted that this modification presents cracks around the periphery, which will be discussed in Section 3.3. The shape of the modifications is distinctly different compared to those in the SP and MHz-burst regimes. One could assume that these modifications are in fact holes of ejected material or voids generated by the intense compression/expansion resulting from the GHz-burst and the Bessel beam shaping method. This assumption will be discussed in the next section.

Figure 4 presents the evolution of the depth of the laser-induced modification throughout the full range of energy for the three regimes. The uncertainty in the depth measurement has been estimated at ±5 µm, and the error bars are not displayed on this graph as they would not be visible. We can infer from this figure that the SP (blue triangles) and the MHz-burst (red diamonds) regimes have a similar behavior as observed in Figure 3. Moreover, a clear trend can be observed from 15 µJ to 55 µJ with the depth increasing at a rate of 5.25 µm/µJ for both regimes. However, an off-set in the initial depth of the modification can be observed between those two regimes. It can be explained by the fact that the energy is distributed among four pulses in the MHz-burst configuration. Thus, the intensity is four times less for an individual pulse and the non-linear absorption is reduced. Consequently, the modifications occur at a higher energy compared to the classical SP regime. This effect is especially pronounced in fused silica, a dielectric with a relatively large band gap (9 eV) [35].

For both the SP and MHz-burst regimes, the energy value of 55 µJ appears to be a threshold beyond which the rate at which the modification depth increases, drops down to 1.45 µm/µJ. We also observe that both regimes can reach a modification longer than 900 µm. This is remarkable considering that, although the Bessel beam is not entirely placed inside the sample, almost 1 mm-deep modifications can be obtained. One can see that we investigated higher energies in the MHz-burst regime than in the SP mode. The reason behind this is that increasing the energy to an equal level in the SP regime would have induced air breakdown and damage to the optical set-up. Lastly, looking at the GHz-burst results (black circles), we observe shorter modifications starting from about 400 µm, and reaching 500 µm for a burst energy of 255 µJ, leading to a rate of 1.19 µm/µJ, which is significantly lower than the rate found for the SP and MHz-burst regimes. In the case of the GHz-burst, the energy is distributed among 50 pulses with a pulse-to-pulse delay of 0.8 ns. Thus, the intensity of each sub-pulse within the burst is much lower compared to the SP and MHz-burst regimes leading to less non-linear propagation effects. Furthermore, the extremely short pulse-to-pulse delay enhances the cooperative effect between the sub-pulses, resulting in a thermal accumulation within the burst in a controlled way. This explains the difference of the GHz-burst regime leading to a direct void instead of internal modifications.

### 3.2. Chemical Etching Results

In this section, we show the results of chemical etching in KOH using the exact same samples with the laser-induced modifications as displayed in the previous subsection. As mentioned in Section 2.3, the samples were etched in a 10 mol·$L^{-1}$ KOH solution for 30 min, 1 h, and 2 h. The microscope images of the samples after 1 h of etching are displayed in Figure 5. All images were taken with the long distance microscope equipped with a 1.4× zoom lens.

Looking at the images, we can distinguish thicker black lines, which can be attributed to the etched regions, whereas the thinner black lines correspond to the bulk modification region, unaffected by the etching process. The etched depths were measured as the lengths of the thicker black lines. Figure 5 shows the evolution of the etching-induced modifications with the energy per SP (Figure 5a) or per burst (MHz-burst, Figure 5b; GHz-burst, Figure 5c). In the SP regime, a slower increase in the modification depth is observed as the energy increases. A slightly larger variance in the modification depths after etching is observed for modifications obtained with the same laser parameters, with respect to the results presented in Figure 3. In the MHz-burst regime (Figure 5b), we observe a clear distinction from the SP regime, with no repeatable etched lines below 55 µJ burst energy. For higher-burst energies, the etching rate increases slowly as in the SP regime. The holes generated by the etching appear wider than in SP. Lastly, Figure 5c presents the etched modifications in the GHz-burst regime. After 1 h, the etching occurs along the full length of the modification and attacks the bulk material outside the laser-induced modifications forming cracks. This observation was also made after 30 min and after 2 h with exactly the same behavior. This demonstrates that the modification produced by a single GHz-burst is indeed a real elongated void reaching the surface and not only a bulk modification. Therefore, the etching process is not a required step in the case of the GHz-burst treatment.

The graph corresponding to the evolution of the etching depth throughout the full range of energies for the different etching times is presented in Figure 6a for the SP regime and in Figure 6b for the MHz-burst regime. The empty symbols (blue triangles for SP and red diamonds for MHz-burst) correspond to the modifications generated by the laser irradiation before chemical etching (values from Figure 3) and are reported here for the sake of comparison. The full symbols correspond to the modifications generated by etching afterward for the different etching times. Results for the GHz regime are not presented since the etching process did not modify further the holes already produced by laser irradiation as previously explained.

In Figure 6a,b, the evolution of the etching depth is similar for both the SP and MHz-burst regimes. In SP regime, after 30 min etching, no effects are observed for pulse energies below 100 µJ. For energies higher than 100 µJ, important deviations are observed within the etched depths of a same group of energy. After 1 h of etching, a clear cut-off can be established. Below 40 µJ of energy per pulse, the etching effect is low and no evident modifications can be observed, a behavior similar to the one recorded after 30 min etching. Above 40 µJ, the etched depth quickly increases to 400 µm and keeps slightly increasing as the pulse energy increases. For energies above 100 µJ per pulse, the data fluctuations are greatly reduced and the etching depth tends to saturate with respect to the energy. This can be confirmed when the etching rates are revealed. For the first hour of etching, a rate of 300 µm/h is reached for 68 µJ and a staggering mean etching rate of 606 µm/h for 128 µJ in SP. However, during the second hour of etching, the etching rate drops down to below 100 µm/h for a pulse energy of 38 µJ and reaches 315 µm/h for 128 µJ highlighting a saturation phenomenon with an etching rate divided by a factor 2 compared to the first hour. When compared to classical etching rates found in the literature, we find an increase for the first hour [30,36] and rather typical etching rates for the 2 h times mark. However, if we take into account our etching conditions, we can assume that our pristine fused silica samples should have an etching rate of around 150 nm/h from [29]. The selectivity represents the improved effect of the etchant on the laser-modified material with respect to the pristine material and is defined as S=(R+ϵ)/ϵ [36], with R representing the laser-modified etching rate and ϵ the pristine material etching rate. In our case, the calculated selectivity is 2103:1 for the 2 h time marks, which, to the best of our knowledge, is the highest reported in the literature.

Concerning the MHz regime, the results are displayed in the graphs of Figure 6b. After 30 min of etching in KOH, similar to the SP regime, we observe no etching. After 1 h, only the modifications generated by a burst energy exceeding 100 µJ show consistent etching to depths varying from 100 µm to 300 µm with fairly high fluctuation from shot-to-shot, presenting more cracks on the edge. The holes are also characterized by a smaller size in comparison to the SP regime, as shown in Figure 5b. Finally, after 2 h in the KOH solution, we observe the same structure as in SP. The etched modifications below 40 µJ burst energy are less visible with depths below 100 µm. Above 40 µJ, the etched depth quickly increases to values ranging from 400 µm to 600 µm for a burst energy of 140 µJ with relatively low fluctuation among the group of same energy. Overall, in the MHz-burst regime, a slight increase in etch depth can be observed. For the first hour, an etching rate of 75 µm/h at 98 µJ is observed, which rapidly increases to 300 µm/h at 105 µJ. Once this value is reached, the etching rate does not fluctuate much, reaching only 315 µm/h at 128 µJ in the MHz-burst regime. However, the MHz-burst configuration differs from the SP regime in the second hour of etching. Here, the etching rate drops below 100 µm/h at a pulse energy of 38 µJ as observed in SP regime, but increases again above 315 µm/h at 113 µJ. Finally, the etching rate increases to a maximum value of 322 µm/hour for 128 µJ burst energy. From this, we can infer that the saturation depth of the MHz-burst configuration has not been reached, as the etching rate for the 2 h is greater than that of the first hour. The selectivity calculated for the modifications obtained in the MHz-burst configuration with 4 ppb is even higher than that obtained in the SP regime, with a value of 2230:1, constituting the highest value reported so far to the best of our knowledge.

To compare the results of the three regimes, the maximum length of the laser-induced modifications, the etching depth, etching rate and selectivity in fused silica for the SP, MHz-burst, and GHz-burst regimes are summarized in Table 1, with the maximum laser-induced modification depth observed for 128 µJ for the SP, and 225 µJ for the MHz-burst and the GHz burst regimes. The maximum etched depths were obtained for 2 h of immersion of the samples in the solution for 100 µJ in the SP and MHz-burst regimes. The maximum etching rate over 1 h and 2 h were obtained for 128 µJ in the SP and for 105 µJ and 128 µJ, respectively, in the MHz-burst regime.

### 3.3. Towards Single GHz-Burst TGV Drilling

As the GHz-burst interaction with the Bessel beam directly creates an elongated void, we investigate in this sub-section the potential for direct, single-step, and dry hole drilling. As detailed in Section 3.1, all modifications were performed with approximately 70% of the Bessel beam inside the sample to ensure a relatively even energy distribution within the sample for the study of etching with respect to pulse energy. More experiments were conducted by placing the entire Bessel beam inside the sample to achieve the longest interaction length within the sample. In this way, we produced longer modifications clearly exceeding the 525 µm length presented in Figure 4. As displayed in Figure 7, we achieved in single shot in pristine fused silica a hole of 730 µm in length with an entrance hole of 4 µm, resulting in an aspect ratio of 182:1. However, it must be noted that this hole presents cracks around the periphery of the affected zone along the hole due to material removal or voids generated by the single GHz-burst. This behavior intrinsically limits the minimum spacing between consecutive holes that can be produced without important damage to the bulk material; hence, this process could be used to generate matrices in bulk material with thicknesses below 700 µm and very tight conduct < 5 µm, with relatively large spacing > 50 µm, depending on the application. Advantageously, the hole generation requires only one step since the hole is generated with a single GHz-burst and without any post-processing etching step. This capability for direct, single GHz-burst drilling in Bessel beam shape paves the way for on-the-fly glass drilling of through-holes as required in TGV.

## 4. Conclusions

Material modifications induced by a femtosecond laser with spatial beam shaping into a Bessel beam have been investigated on fused silica in different temporal beam shaping regimes, including the standard single-pulse mode, the MHz-burst mode containing 4 pulses per burst at 40 MHz repetition rate, and the GHz-burst containing 50 pulses per burst at 1.28 GHz repetition rate. All three regimes have been applied in single shot—pulse or burst—operation mode. The pulse or burst energy has been varied and compared across all three regimes. This study revealed that changing from the single-pulse regime to the MHz-burst regime does not significantly affect the morphology and quality of the modifications in the glass. In contrast, the GHz-burst regime drastically affects the shape and leads to a different type of elongated microstructures, while single pulses and short MHz-bursts result in deeper modifications, the GHz-burst regime directly produces a uniform channel with a single burst.

Furthermore, we investigated the influence of the pulse/burst energy on the chemical etching rate in these three different regimes for various times of immersion. This study showed that the single pulse regime presents a higher etching rate for the first hour with 606 µm/h before reaching a plateau during the second hour where the etching rate gets halved. Nevertheless this single pulse regime at high energy revealed a remarkably high selectivity of 2103:1 in fused silica. Therefore, this regime is preferable for limited depth drilling with a fast process. On the contrary, the MHz-burst regime presents the lowest etching rate in the first hour with at maximum 315 µm/h but does not reach the same plateau as the maximal etching rate reached after two hours is 322 µm/h corresponding to only a slight increase. Moreover, the MHz-burst regime features an even higher selectivity of 2230:1, which is the highest ever reported so far to the best of our knowledge. This regime allows to obtain the greatest hole depths and is thus preferable for applications requiring a maximum hole depth. Precise control over the laser parameters allowed us to have a perfect mastery of the hole depth and morphology.

Finally, the GHz-burst regime offers a new, chemical-free, single step and fast approach to fine hole drilling. It could be implemented in on-the-fly glass drilling as a unique solution, since it presents a noticeably shorter processing time than the single-pulse and MHz-burst regimes, respectively. This is due to only one single burst being required to generate a hole without any post-processing etching steps. However, this new technique suffers from some quality defects with the appearance of some cracks. Overall, this study constitutes a significant contribution to the collective understanding of laser processing using the GHz-burst regime as well as the behavior of MHz-bursts with respect to chemical etching, and holds interesting potential for industrial precision micromachining of dielectric materials.

## Figures and Tables

**Figure 1 micromachines-15-01313-f001:**
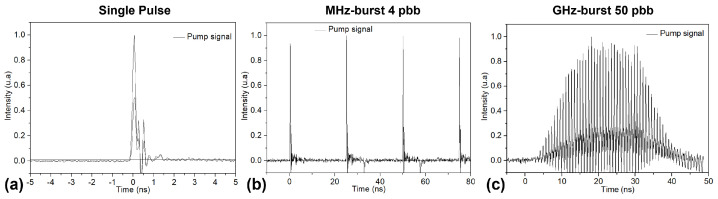
Oscillograms corresponding to the single-pulse regime (**a**), the MHz-burst regime with 4 pulses per burst (**b**) and the GHz-burst regime with 50 pulses per burst (**c**), respectively.

**Figure 2 micromachines-15-01313-f002:**
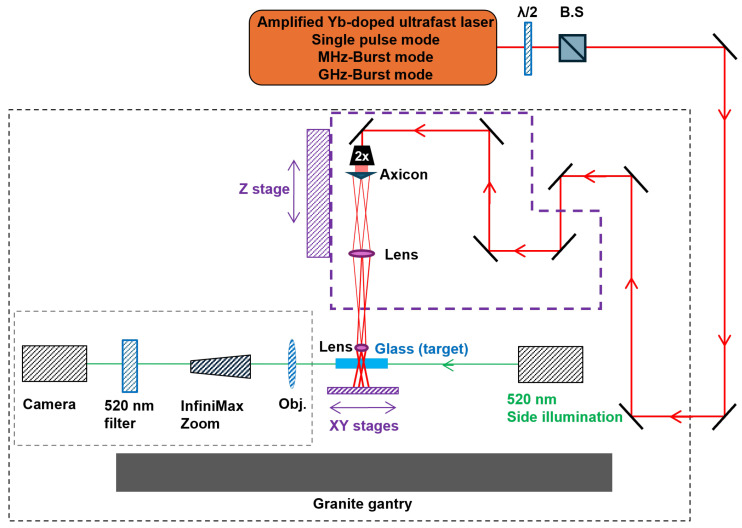
Experimental setup used for Bessel beam irradiation in fused silica.

**Figure 3 micromachines-15-01313-f003:**
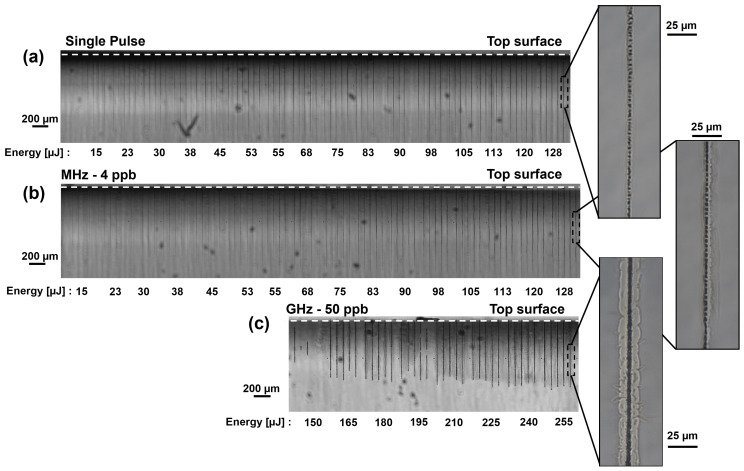
Optical microscope side views of modifications obtained in fused silica after laser irradiation with a single pulse or burst, respectively, at an energy ranging from 15 to 255 µJ, (**a**) Single-pulse mode, (**b**) 4 ppb MHz-burst and (**c**) 50 ppb GHz-burst. The pitch between two modifications of same energy is 50 µm. The white line at the top of each microscope image represents the top surface. The inserts on the right are a zoom of the highest energy modification delimited by a rectangle of black lines.

**Figure 4 micromachines-15-01313-f004:**
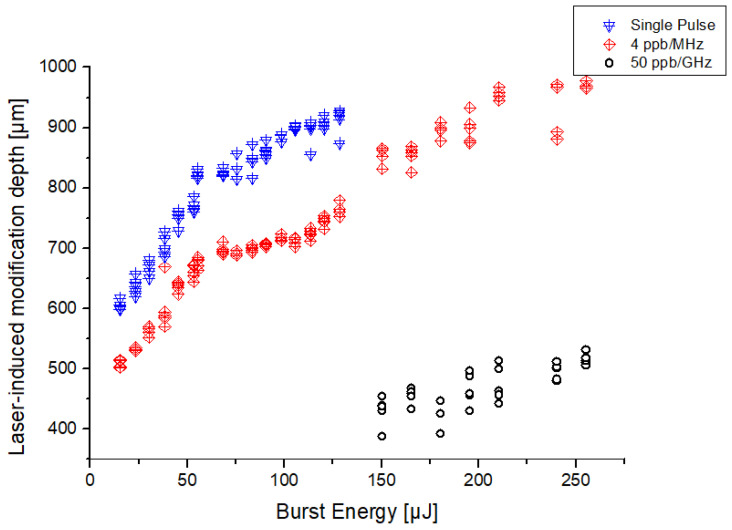
Evolution of the modification depth obtained in fused silica with a single pulse/burst as a function of energy in the range from 15 to 255 µJ, in single-pulse mode, 4 ppb MHz-burst, and 50 ppb GHz-burst. For each energy, at least 4 modifications were measured.

**Figure 5 micromachines-15-01313-f005:**
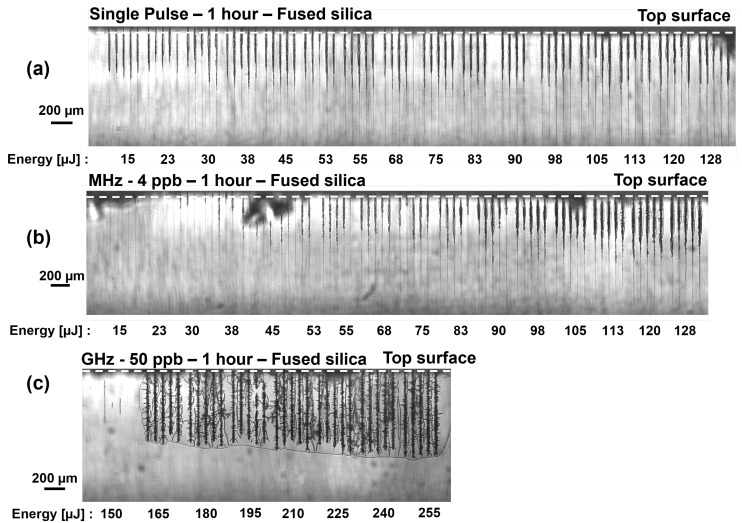
Optical microscope images of the modifications obtained in fused silica after laser irradiation with one single pulse or burst, respectively, with an energy ranging from 15 to 255 μJ, in (**a**) Single pulse mode, (**b**) 4 ppb MHz-burst, and (**c**) 50 ppb GHz-burst and after 1 h of KOH etching.

**Figure 6 micromachines-15-01313-f006:**
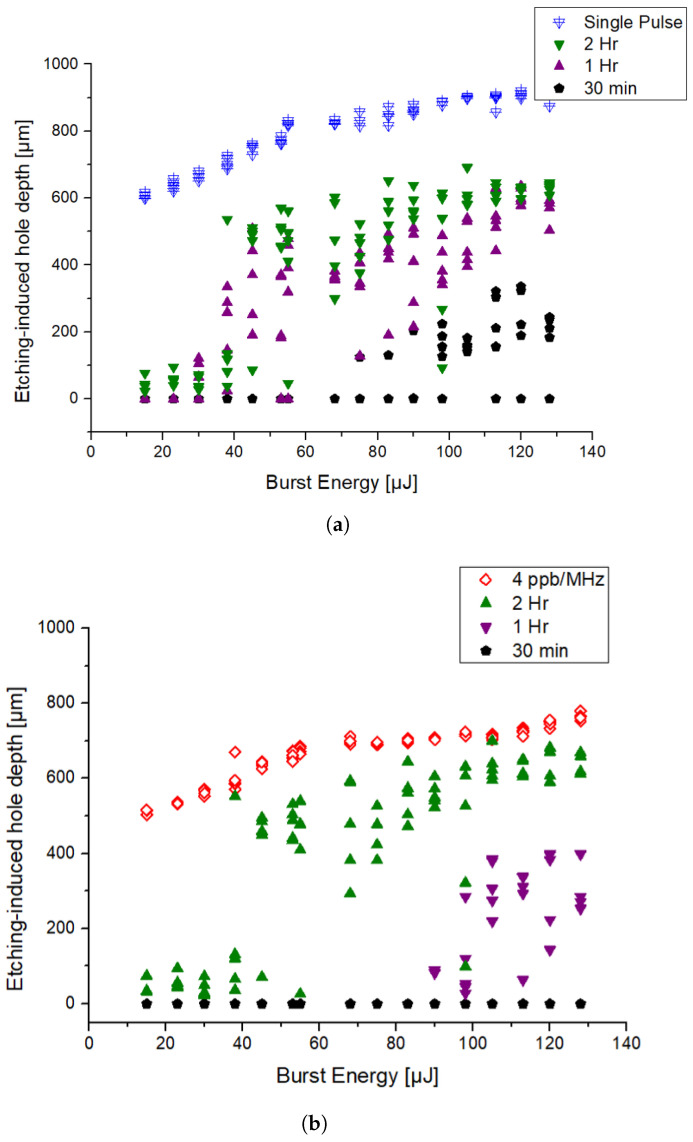
(**a**) Evolution of the modification depth obtained in fused silica after a single pulse (before chemical etching) and of the hole depth after chemical etching as a function of the pulse energy, ranging from 15 to 140 µJ, for different etching times. (**b**) Evolution of the modification depth obtained in fused silica after a single MHz-burst (before chemical etching) and of the hole depth after chemical etching as a function of the burst energy, ranging from 15 to 140 µJ, for different etching times.

**Figure 7 micromachines-15-01313-f007:**
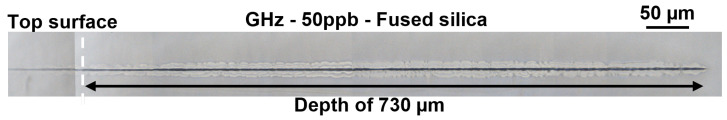
Void microstructure obtained in fused silica with one single GHz-burst (50 ppb) of 255 µJ without chemical etching. Top surface is on the left.

**Table 1 micromachines-15-01313-t001:** Maximum achievable laser-induced modification, etched depth, etching rate, and selectivity in fused silica for the SP, MHz-burst, and GHz-burst regimes.

	Max. Laser-Induced Modification Depth	Max. Etching Depth	Max. Etching Rate over 1 h	Max. Etching Rate over 2 h	Selectivity
	**[µm]**	**[µm]**	**[µm/h]**	**[µm/h]**	
SP regime	930, at 128 µJ	650 at 100 µJ	606, at 128 µJ	315, at 128 µJ	2103:1
MHz-burst regime	950, at 225 µJ	680 at 100 µJ	300, at 105 µJ	322, at 128 µJ	2230:1
GHz-burst regime	525, at 225 µJ	/	/	/	/

## Data Availability

Data underlying the results presented in this paper are not publicly available at this time but may be obtained from the authors upon reasonable request.

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
