# Peer review of "Bessel Beam Femtosecond Laser Interaction with Fused Silica Before and After Chemical Etching: Comparison of Single Pulse, MHz-Burst, and GHz-Burst"

_micromachines, 2024, doi:10.3390/mi15111313_

Round 1
Reviewer 1 Report
Comments and Suggestions for Authors
Please find my report in attachment.

I would suggest a cross check by an english mothertongue speaker.
Reviewer 2 Report
Comments and Suggestions for Authors
The paper “Bessel beam Femtosecond Laser Interaction with Fused Silica before and after Chemical Etching : Comparison of Single Pulse, MHz-burst, and GHz-burst” by Théo Guilberteau with coauthors is devoted to studies of peculiarity of holes drilling in SO2. In particular it is shown, that GHz repetition provide much better quality of the hole drilling. Even more important, that after GHz burst the holes in the SiO2 glass are physically created and does not require additional chemical etching. In general, I really like this paper, but suggest that it could be improved if authors address the following comments:
1. Fig2c_what is the light stripe on both sides of the hole - is it modified material?
2. What was repetition rate of single-pulse, MHz-burst and GHz burst?
3. How each hole was drilled? I mean how many bursts were passed to each hole? Was z stage was moved during hole drilling? If yes - which distance and was step per burst?
4. How the pulse energy was changed? From the paper it is looks like that step-like (~ 4 holes were drilled at the same energy), but it is quite natural designate at the image those 4 holes was drilled at the indicated energy. Or energy change was continuous-like (each new hole was drilled with increased energy)?
5. It is highly desirable to add into the text explanation for KOH (abstract and line 52) – to make it clear that it is chemical formular for Potassium Hydroxide and nor abbreviation, which authors forgot to explain. Similar ”10 M KOH etchant solution” (line 101) must be written more clearly – it is not obvious what does it mean “10 M” for non-specialists.
